# Nutritional Therapy in Pediatric Crohn’s Disease—Are We Going to Change the Guidelines?

**DOI:** 10.3390/jcm10143027

**Published:** 2021-07-07

**Authors:** Malgorzata Matuszczyk, Jaroslaw Kierkus

**Affiliations:** Department of Gastroenterology, Hepatology, Feeding Disorders and Pediatrics, The Childrens’ Memorial Health Institute, 04-730 Warsaw, Poland; j.kierkus@ipczd.pl

**Keywords:** Crohn’s disease, guidelines, induction of remission, nutritional treatment, elimination diet

## Abstract

In recent years, there has been a significant increase in the incidence of Crohn’s disease. Despite significant medical progress, the treatment options available today do not meet the needs of all patients. Recent reports indicate that external environmental factors, including diet, are key in the pathomechanism of the disease. It was proven that the so-called Western dietary pattern is associated with an increased risk of disease. In the pediatric population, exclusive enteral nutrition is the only nutritional therapy option recommended today with proven high efficacy in inducing remission. Recent publications that indicate at least comparable efficacy and significantly better tolerability of a specialised elimination diet, the Crohn’s Disease Exclusion Diet (CDED), provide the basis for a change in recommendations. This article discusses the mechanism of action, principles of use, and scientific evidence evaluating the efficacy of CDED in the treatment of children with Crohn’s disease.

## 1. Introduction

The incidence of Crohn’s disease (CD), a chronic inflammatory disease of the gastrointestinal tract, has increased dramatically over the past two decades [1]. Increasingly, this problem affects children—according to current data, approximately 10% of patients have a diagnosis before the age of 17 [2]. Despite significant progress in the field of pharmacological treatment, its effectiveness is still unsatisfactory—at present, we do not have a drug that would act not only on the effects but also on the causes of the disorders. CD remains an incurable condition that is difficult to control and has a risk of recurrence and complications. The progressive nature of the disease, resistance/loss of response to subsequent drugs, the need for surgical treatment, or the use of steroids and immunosuppressive drugs characterised by a high profile of side effects, further worsen the prognosis [3,4,5].

Recent data suggest that the development of chronic inflammation in CD may be related to specific external environmental factors. One of the key factors that can negatively impact the delicate immune balance between the microbiome and the intestinal mucosa is diet [6,7,8]. This concept is supported by the results of studies in cellular and animal models as well as epidemiological studies, which indicate a positive correlation between the so-called “Western” dietary pattern and increased risk of CD [6,9,10]. These reports gave rise to a growing interest in nutritional therapy as an alternative therapeutic option to pharmacological treatment.

The primary goal of treatment for CD is to achieve deep remission, i.e., clinical, biochemical and histopathological remission [11,12]. In the pediatric population, it was proved that exclusive enteral nutrition (EEN) is more effective in inducing deep remission than systemic steroids [13,14]. In addition, this form of treatment has a beneficial effect on the nutritional status and bone mineral density, often impaired by the disease, and is free of side effects. For these reasons, EEN is recommended today as the first-line treatment for active CD in children [5,15,16]. However, nutritional therapy, which consists of a complete exclusion of the natural diet and feeding solely on a specialised preparation (enteral diet) for 6–8 weeks, is difficult in practical application and lacks the concept of long-term management to maintain the therapeutic effect [17,18].

The mechanism of action of EEN is to exclude ingredients from the diet that negatively affect the homeostasis between the microbiota, the intestinal mucosa and the immune system [6,7]. The lack of a more affordable option for nutritional therapy has, until recently, been a barrier to large-scale use of this treatment. Previously published data on the therapeutic efficacy of specialised elimination diets (e.g., specific carbohydrate diet or autoimmune protocol diet) were not groundbreaking and did not indicate that EEN could be replaced by a diet composed of appropriately selected natural products [16,19]. However, in 2019, results of a multicenter randomised trial were published, indicating comparable efficacy of EEN and a specific elimination diet developed for patients with CD, the CDED, in inducing remission in children with active CD. Importantly, therapy tolerance was significantly higher in the CDED group than in the EEN group [7]. The results of the study and regularly published summaries of subsequent analyses give hope that this modern method of nutritional treatment will soon find its place in official recommendations, replacing or providing an alternative to EEN. The purpose of this article is to discuss the mechanism of action and protocol of the CDED diet as well as the results of scientific reports supporting its efficacy in the treatment of CD in the pediatric population.

## 2. Diet and the Pathogenesis of Crohn’s Disease

The etiology of CD is unknown. Currently, interactions between environmental factors and the intestinal microflora and immune system (intestinal barrier) in individuals with a genetic predisposition to develop the disease are considered the most likely pathomechanism (Figure 1) [20]. The critical importance of environmental factors is confirmed by epidemiological data, including incidence rates around the world. Interesting information was provided by observations among people who moved from regions with low incidence to countries with high incidence—it was shown that the offspring of immigrants had the same risk of CD as children coming from families living in regions with high incidence for many generations [21]. One of the best-studied environmental factors in the context of CD pathogenesis is diet. Recent data suggest that as a result of specific dietary factors, the delicate immune balance between the microbiota and the intestinal mucosa is disrupted, leading to chronic inflammation. This hypothesis is supported by research findings that show a correlation between the so-called Western (industrialised) dietary pattern and increased risk of CD [6,9,10].

Epidemiological studies proved the influence of the Western diet on an increased incidence of CD and, conversely, a lower incidence among those following a Mediterranean diet [6,10]. One of the features typical of the Western dietary pattern is the high consumption of processed foods. Meanwhile, a number of adverse consequences resulting from exposure to a variety of food additives were demonstrated in studies in cellular and animal models [6]. For example, the destructive effects of emulsifiers (i.e., carboxymethylcellulose and polysorbate-80) on the mucus layer that protects intestinal epithelial cells was proved [22,23,24]. Due to their gelling and thickening properties, these additives are commonly used in meat and dairy products, among other things. Another problem specific to the Western diet is low dietary fibre intake. Particularly relevant to the pathogenesis of CD may be a low supply of food that provides substrates for the production of short-chain fatty acids (SCFAs), especially water-soluble dietary fibre and resistant starch [8]. Short-chain fatty acids, which are a metabolic product of the bacteria residing in the intestines, play an important role in maintaining normal intestinal barrier and immune system function in a number of ways, including:by influencing the activity of immune cells and their migration to the site of inflammation, they exhibit anti-inflammatory effects;by nourishing colonocytes, providing them with a primary source of energy;by reducing the pH in the intestine, positively influencing the composition of the intestinal microbiota (stimulating the growth of beneficial strains of bacteria and inhibiting the growth of pathogenic bacteria).

On the other hand, in the absence of sources for SCFA production, the mucus layer that protects intestinal epithelial cells can be used by bacteria as a medium, allowing their translocation through the intestinal mucosa [8,25].

Other features of the Western diet with potential negative effects on CD development include high consumption of red/processed meat, animal fat, and wheat (Figure 2) [6]. For example, studies in animal models proved, among other things, the effect of a diet high in fat and sugars on adverse changes in the composition of the intestinal microbiota and on impaired expression of short-chain fatty acid receptors (GPR43). GPR43 are involved in mechanisms regulating SCFA-mediated immune responses [26].

A summary of Western dietary factors considered as particularly important in the pathomechanism of inflammation in CD is shown in Figure 2.

## 3. Why Are the Currently Recommended Nutritional Treatments in the Pediatric Population Not Fully Satisfying for Us?

The achievement of deep remission with healing of the intestinal mucosa has a fundamental impact on long-term treatment outcomes and is an essential goal of CD treatment today [12]. In the pediatric population, the comparable efficacy of EEN and glucocorticosteroids in the treatment of active CD, with a significantly more favorable effect of nutritional therapy on intestinal mucosal healing, was sufficiently proved in high-quality scientific studies [5,27]. Moreover, unlike steroid therapy, nutritional treatment has a positive effect on nutritional status and is devoid of side effects [15]. Therefore, exclusive enteral nutrition is now recommended as the first-line treatment for the active luminal CD in children [16,28]. The implementation of EEN into daily clinical practice enabled a significant reduction in the use of corticosteroids in the pediatric population. However, the therapy, which requires complete exclusion of the natural diet and feeding solely on a specialised formula (enteral diet) for 6–8 weeks, is difficult to implement in practice—it requires high motivation from the patient and parents, and in some cases, it requires the use of a nasogastric tube [17,18]. In addition, this idea of nutritional treatment lacks a management strategy to maintain remission and the rate of disease re-exacerbation after returning to the habitual diet is high [18,28,29].

EEN is a safe, effective, and so far, the only causal treatment concept that achieves clinical and endoscopic remission in the majority of treated children (Figure 1) [5,15,16,27,28]. The previously mentioned limitations of EEN are an obstacle to the large-scale application of this therapeutic method. Until recently, data on the possibility of application of different elimination diets have not been promising and no nutritional treatment concept other than EEN has found its way into the official recommendations of major scientific societies so far [17]. However, over the past few years, further evidence has emerged for the effectiveness of a specialised elimination diet developed for CDED, in treating children with active CD [7,30,31]. These reports are very promising and offer hope for replacing EEN, which is cumbersome to apply, with this innovative method of nutritional treatment.

## 4. CDED—Protocol and Mechanism of Action

The CDED diet is a new generation of nutritional therapy—in its initial stages, aimed at inducing remission, it involves a combination of partial enteral nutrition (PEN) with selected natural diet products. The primary mechanism of action is to exclude or limit exposure to dietary factors with potentially deleterious effects on the pathogenesis and course of CD (Table 1).

The direct effect of the elimination diet on restoring eubiosis, normal intestinal barrier function, and immune response, causes the quenching of inflammation and healing of the intestinal mucosa (Figure 3).

The auxiliary mechanism of CDED involves consideration of the supply of specific components that may provide additional benefits [7,30,31]. A key role in this regard is attributed to the effect of increasing the production of short-chain fatty acids. Therefore, foods that are mandatory in CDED include selected foods rich in water-soluble fibre (e.g., pectin) as well as resistant starch (Table 2) [6,7,30,31].

In the first phase, due to the greatest restrictions regarding the allowed foods, 50% of daily energy demand must be satisfied by enteral diet. Phase 2 is the time of reintroduction of some foods that had to be eliminated during the first 6 weeks. Due to the increasing variety of foods that are allowed, the recommended % of energy provided with the formula is reduced to 25% of the demand. Satisfying some nutritional needs with the use of a complete enteral diet is aimed at preventing deficiencies and maintaining/improving the nutritional status in the period of the greatest restrictions, i.e., in the first 2 phases of the diet. The principles of formula selection are the same as the guidelines for exclusive enteral nutrition in children. Standard polymeric, normocaloric (1 kcal/1 mL) diets are recommended. In justified cases, that is in patients with food intolerances, allergy to cow’s milk proteins or insufficient tolerance of the polymeric preparation, preparations containing hydrolysed protein should be recommended.

Foods that can be consumed in the first two phases of the CDED are divided into foods that are:

–mandatory, i.e., recommended for daily consumption. Their role is to provide adequate nutritional value of the diet as well as substrates for the production of SCFA;–neutral, which are supposed to add variety to the daily menu, but do not necessarily have to be consumed.–forbidden.

Table 2 presents the summary of mandatory and neutral products that can be consumed during phases 1 and 2 of the diet. The quantity of enteral feeding and particular supplementing products should be determined individually, in accordance with the state of nutrition stemming from sex, age and activity of the disease as well as energy and nutrient demand.

In phase 3 (maintenance), patients should function according to the principle of controlled exposure to dietary components with potentially negative effects on pathogenesis and the course of the disease. Therefore, for five days a week, they should compose their meals based on products allowed in phase 2 as well as selected additional products. During these days, especially in the case of patients who did not normalise the parameters of their nutritional status despite good treatment effects, it is recommended that supplementation with formula be continued. In addition, on selected consecutive days it is possible to eat two meals composed of products that are not recommended for daily consumption. Patients should still avoid particularly harmful foods, mainly highly processed products, i.e., processed meats, frozen, ready-to-eat foods and sweetened beverages. Table 3 presents a summary of foods/meals that can be consumed additionally during the maintenance phase of CDED.

## 5. Effectiveness of Crohn’s Disease Exclusion Diet in Studies

CDED was developed in 2010 by Professor Arie Levine (Wolfson Medical Center, Tel Aviv). The first reports—results of a retrospective analysis of treatment results of CDED followed for 12 weeks by a group of 47 children and young adults with active CD—were published in 2014 [30]. After the end of the first phase of CDED, i.e., after six weeks, clinical remission was achieved in 33/47 patients (70.2%) in the study group. Intestinal mucosa healed in 11 (70%) out of 15 patients who underwent endoscopic examination before the beginning of the diet and after following it for six weeks. A publication confirming high efficacy of CDED in a group of 21 patients who lost response to biological treatment was published in 2017 [31]. After 6 weeks, 62% of patients were in clinical remission, and in 38% and 43% of patients, respectively, researchers observed normalisation or a decrease in the parameters of inflammatory state. In 2019, Gastroenterology published groundbreaking results of a multicentre randomised study, indicating that the effectiveness of CDED and standard treatment (EEN) in terms of the induction of clinical remission in children with active CD (80% vs. 73.5% in the sixth week of treatment, respectively) was comparable, and that the treatment of significantly better tolerated in the intervention group [7]. However, after 12 weeks of observation, 76.2% of children treated with CDED remained in steroid-free remission, compared to only 45.1% of patients in the EEN group. In addition, patients treated with the new exclusion diet showed better results with regard to reducing intestinal permeability. After 3 weeks of nutritional treatment the intestinal permeability lowered in the CDED group while increased in the EEN group. What is more, both in the CDED and EEN groups the correction of dysbiosis was observed at week 6 (decreased proteo-bacteria), however, the reduction in proteobacteria was maintained at 12 weeks only in the CDED group, while a major rebound of this tax was observed in EEN group at the same time point. Unlike in EEN, the beneficial effect of CDED on the reduction of the intestinal permeability and on the favorable and persistent modification of microbiota composition may be of key importance for long-term maintenance of treatment effects. In the past year, researchers published a presentation of a case series, suggesting that the range of indications for CDED may potentially be broader [32]. The paper confirmed high effectiveness of CDED, applied not only as monotherapy, but also within the framework of a therapy combined with pharmacological treatment and as a salvage treatment in patients resistant to pharmacological treatment. Further studies are underway or at the stage of data analysis, including a publication presenting the results of a randomised study evaluating the efficacy of CDED compared to standard treatment in adult patients.

## 6. The ModuLife Project and Application

In 2019, the ModuLife software, developed by the creators of CDED in collaboration with Nestle, was made globally available for use. The primary goal of the project is to train CDED specialists and popularise the method among patients. The application for patients contains a recipe/meal database and provides a lot of help in everyday diet. Access to the platform can only be granted by a CDED expert. The training for specialists is available at: https://modulifexpert.com/ (accessed on 6 July 2021). Apart from lectures, it includes auxiliary materials as well as regularly held seminars that supplement the latest knowledge on CDED [33].

## 7. Summary

As of today, EEN is the sole method of nutritional treatment with proven efficacy in inducing remission in children with active Crohn’s disease that is recommended in the official guidelines. This form of dietary treatment, however, has significant limitations that have a negative impact on the possibility of its wide use and long-term maintenance of its positive effects. It seems that the groundbreaking reports concerning CDED, a modern method of dietary treatment, which indicates that it is at least comparable with EEN in terms of its effectiveness in inducing remission and that is much better tolerated by patients, CDED will be included in the latest guidelines of scientific societies.

## Figures and Tables

**Figure 1 jcm-10-03027-f001:**
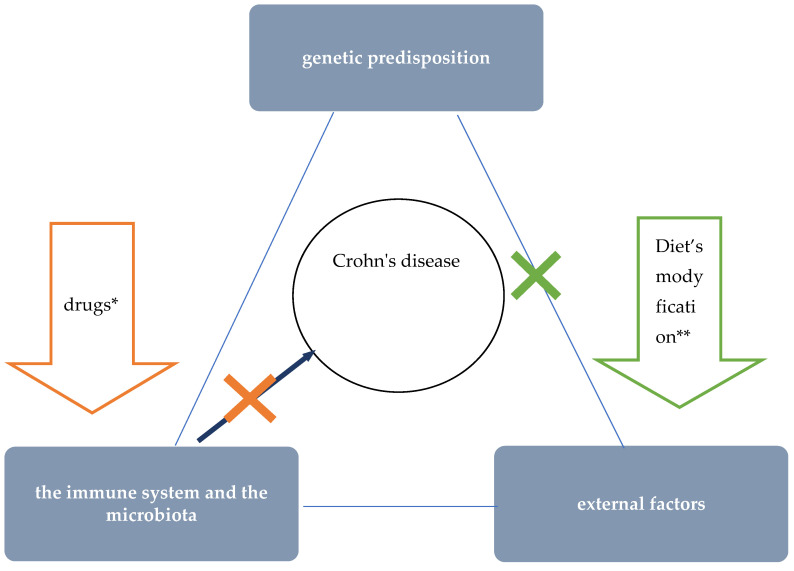
The role of nutritional therapy in the treatment of Crohn’s disease. * steroids, immunosuppressants, and biologics’ ** such as EEN.

**Figure 2 jcm-10-03027-f002:**
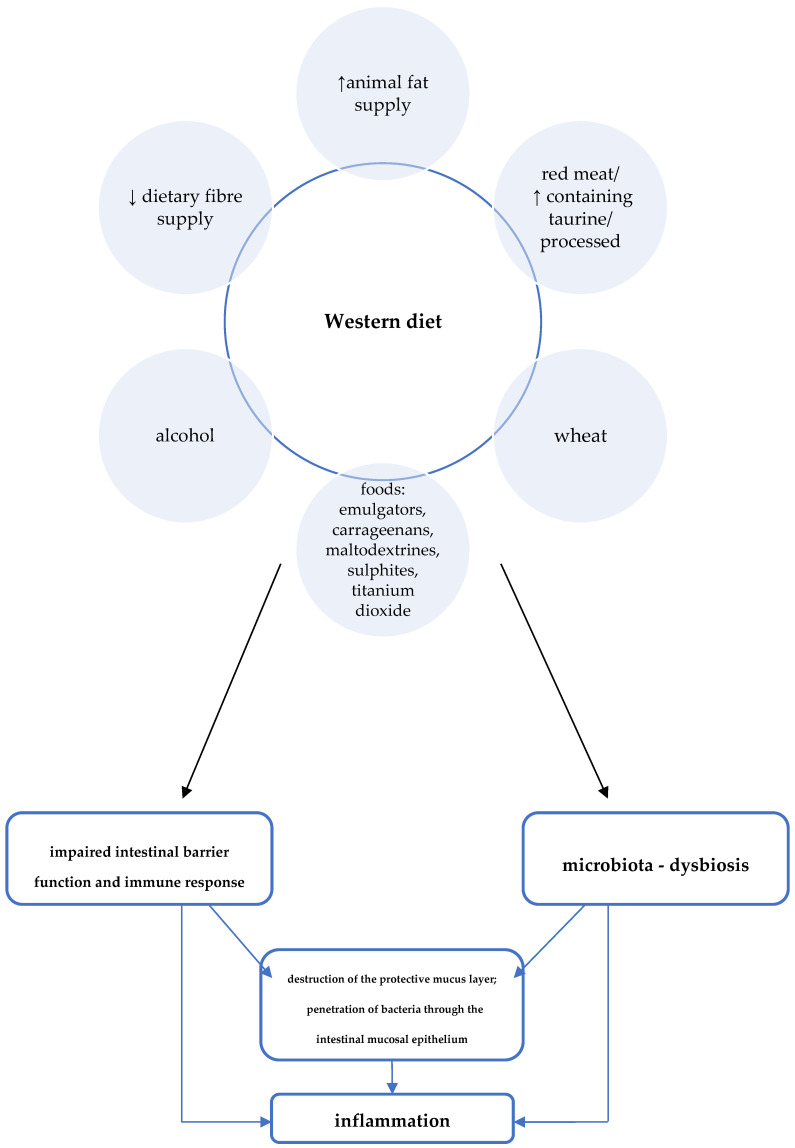
Western diet and the pathogenesis of Crohn’s disease [6].

**Figure 3 jcm-10-03027-f003:**
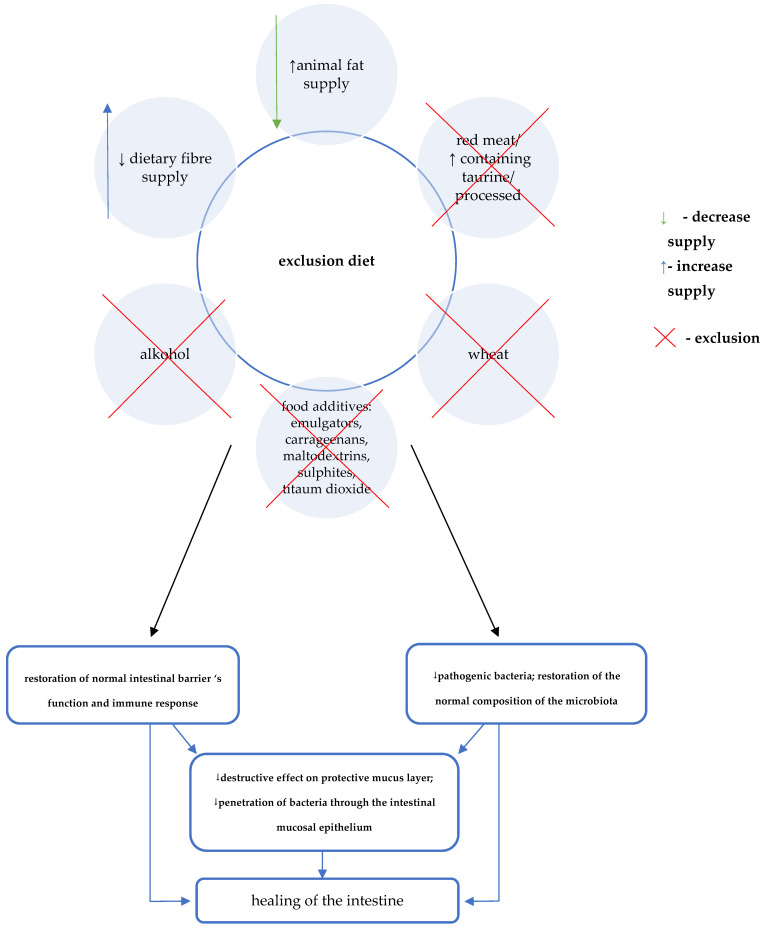
Crohn’s Disease Exclusion Diet (CDED) diet and the treatment of Crohn’s disease [7,30,31].

**Table 1 jcm-10-03027-t001:** Major food groups eliminated in the Crohn’s Disease Exclusion Diet.

Natural Products—Exclusion or Controlled Exposure	Food Additives—Exclusion
products high in animal fat; products rich in taurine, i.e., fish; offal; red meat; dairy products (rich in animal fat and sulfate-reducing bacteria); wheat (contains amylase and trypsin inhibitors as well as gluten); water-insoluble fibre (quantitative limits) alcohol yeast (affects dysbiosis)	emulsifiers (dairy products; sauces; spreads; frozen baked goods, low-fat products) carrageenans (cheese, dairy products; beer) maltodextrins (breakfast cereals; powdered drinks; artificial sweeteners) sulfites (dried fruit; wine; wine vinegar; processed fruit; canned fruit; frozen vegetables) titanium dioxide (chewing gum; powdered sugar, processed tahini paste)

**Table 2 jcm-10-03027-t002:** Foods mandatory/allowed during the induction phase of CDED (stages 1 and 2).

Mandatory Foods (Stages 1 and 2)		
**A balanced diet** **—** **source of complete protein with low taurine content**		**Additional benefits** **—** **sources of water-soluble dietary fibre (pectins) and resistant starch**
chicken breast eggs		potatoes * (resistant starch) bananas ** (resistant starch) apples (pectin)
**Group of foods**	**allowed in stage 1** **(weeks 1** **—** **6)**	**Additionally allowed in stage 1 (weeks 1** **—** **6) *****
Cereals	white rice, rice flour and rice pasta (in unlimited amounts)	- quinoa (in unlimited amounts) - sweet potato (½ can replace 1 potato, once a day) - oatmeal—½ cup 1—2 times a week (can be used to make oatmeal or oatmeal cookies) - one slice of whole-grain bread/day (without yeast)
Meat/fish/eggs	instead of chicken breast, a serving of lean white fish once a week.	- fresh lean beef (lean meat i.e., sirloin) can be eaten once a week instead of chicken breast - a serving of tuna in canola oil or olive oil once a week
Vegetables	- 2 tomatoes or 6 cherry tomatoes - 2 peeled cucumbers - 1 young carrot - fresh spinach (a cup once a day) - lettuce leaves (3 per day) - avocado (1 a day; max. ½/meal)	gradual introduction of new vegetables: - initially those containing lower amounts of dietary fibre (e.g., 1 small zucchini, 2 broccoli or cauliflower florets) - starting from week 10 other vegetables can br introduced (except for cabbage, leak, asparagus, artichoke and celery)—e.g., ½ of sweet red bell pepper, beet
Fruits	- strawberries (several a day) - a slice of melon	gradual introduction of new fruit: in weeks 7–9 1 pear, peach or kiwi/a day can be eaten—instead of a serving of strawberries, 10 bilberries (or a cup) can be eaten - form week 10 other fruits can be introduced (which contain more fibre), but in limited quantities—i.e., ½ cup of mango, pineapple cubes or orange slices (except for passion fruit, pomegranate, cactus, kaki)
Fats	- olive oil - canola oil	
Sugar and sweets	- honey (3 teaspoons a day) or - sugar (4 teaspoons a day)	
other		- legumes (lentils, chickpeas, beans, peas)—½ cup dry seeds a day) - almonds or walnuts (unsalted, unroasted, unprocessed)—8 pieces a day - tahini (without emulsifying agents and sulphites)—2 spoons a day
fluids	- water (slices of lemon, lime, orange or mint leaves can be added to taste) - herbal teas (preferable from fresh leaves) - 1 glass of freshly squeezed orange juice	
Spices	-salt, pepper, paprika, cinnamon, cumin, turmeric - fresh herbs i.e.: mint, oregano, cilantro, rosemary, sage, basil, thyme, dill, parsley - other spices: onion, garlic, ginger, fresh lemon juice	

* preferably boiled and cooled. ** preferably medium mature. *** as well as all foods allowed during stage 1. The protocol of the CDED is divided into three phases, including 2 (lasting 6 weeks each) stages of the induction stage and phase 3—maintenance, which should be continued for at least 9 months, and treated as the target diet. Each subsequent phase is less restrictive and easier.

**Table 3 jcm-10-03027-t003:** Foods allowed in the maintenance phase of CDED.

Foods Allowed 5 Days a Week *	Examples of Products Allowed 2 Days a Week **	
	Examples of Foods That Can Be Consumed for Breakfast:	Examples of Foods That Can Be Consumed for Lunch (Dinner):
- other parts of chicken (except for skin or giblets) - fresh seafood, salmon (once a week)—1 serving of unprocessed, full-fat natural yoghurt (without additives) a day - 2 slices of whole-grain bread (without yeast), or 1 serving of pasta a day - all vegetables except for leak, celery stalks and large amounts of kale - all fruits (including dried fruits, without sulphites) except for passion fruit, pomegranate, cactus, kaki - 1 cup of black coffee (not instant) or tea	- any kind of bread - milk, cheese - crêpes - jams - 1 bowl of cereal with milk	- steaks, burgers, pork, seafood, fish, i.e., salmon and tuna, - any type of pasta - dairy products, including cheese - 1 serving of home-made dessert (e.g., cake) or one scoop of ice-cream - cocoa/dark chocolate
	- or 1 meal eaten out instead of unrestricted meals prepared at home	

* as well as all foods allowed in phase 2. ** can be eaten for 2 selected meals during the day.

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
