# Peer review of "Nutritional Therapy in Pediatric Crohn’s Disease—Are We Going to Change the Guidelines?"

_jcm, 2021, doi:10.3390/jcm10143027_

Round 1
Reviewer 1 Report
In this review, the authors have summarized the most recent updates concerning Crohn’s disease exclusive diet (CDED) and they discuss its implementation protocol and its potential to replace EEN as a first line therapy. This review represents a good contribution for the field of Crohn’s disease therapy. The authors have added very nice tables summarizing some nice information about CDED.
I have some comment that may help the authors to improve the readability.
- I suggest that you add some text (text legend) underneath of the figures to explain the flow of the story of your figures.
- Figure 2: I cannot fellow, what are the green and red crosses? Why different colors? You are talking about the role of nutritional therapy, but I cannot see any cartoon or sign about nutritional therapy in your illustration. I think is better to add some text to explain the figure.
- Figure 3: what do you mean by elimination diet? You mean exclusion diet? I think its better to use same terminology you are using in the text. I see a red cross near to the word exclusion and green a bleu row near to word supply. I mean they are a legend referring to an effect in your figure, however I do not see any cross or green/bleu arrows in your figure. Adding text explaining your figure can be useful for the readers.
- Line 179, you already mentioned the abbreviation of CDED in the intro, so you can remove Crohn’s disease exclusion diet and keep just CDED.
- The authors can discuss more the difference between CDED and EEN in term of clinical remission and sustained remission. In the intro, the authors have mentioned that disturbance microbiota is one the main factors contributing to the development of CD. However, in the text, it was neglected. May the author can discuss that a bit. For example, Both CDED and EEN were able to induce a same rate of remission. However, CDED and not EEN was able to sustain remission. Arie Levine et al, have showed that both CDED and EEN were able to correct dysbiosis by week 6 (decreased proteobacteria), however only CDED was able to sustain the reduction on proteobacteria till W12, while a major rebound of this text was observed in EEN at week 12. the reader can have a great idea about the differences between CDED and EEN.
Author Response
Comment 1: Figure 2: I cannot fellow, what are the green and red crosses? Why different colors? You are talking about the role of nutritional therapy, but I cannot see any cartoon or sign about nutritional therapy in your illustration. I think is better to add some text to explain the figure.
Authors' response. Thank you for the very valuable comment. The Figure haas been corrected according to sugestions. Additionally, a reference to the figure was added in the text (lines 69-72) explaining the current concept of the pathomechanism of the disease. The colors of the crosses have been distinguished to show that the green cross refers to the action of the diet's modyfication marked in green on the figure, and the orange cross refers to the action of the drugs marked in orange on the figure.
Comment 2: Figure 3: what do you mean by elimination diet? You mean exclusion diet? I think its better to use same terminology you are using in the text. I see a red cross near to the word exclusion and green a bleu row near to word supply. I mean they are a legend referring to an effect in your figure, however I do not see any cross or green/bleu arrows in your figure. Adding text explaining your figure can be useful for the readers.
Authors' response: Thank you for the very valuable comment. Figure 3 has been corrected according to suggestios.
comment 3: Line 179, you already mentioned the abbreviation of CDED in the intro, so you can remove Crohn’s disease exclusion diet and keep just CDED.
Authors' response- Thank you for the comments. diet's name updated to CDED.
comment 4: The authors can discuss more the difference between CDED and EEN in term of clinical remission and sustained remission. In the intro, the authors have mentioned that disturbance microbiota is one the main factors contributing to the development of CD. However, in the text, it was neglected. May the author can discuss that a bit. For example, Both CDED and EEN were able to induce a same rate of remission. However, CDED and not EEN was able to sustain remission. Arie Levine et al, have showed that both CDED and EEN were able to correct dysbiosis by week 6 (decreased proteobacteria), however only CDED was able to sustain the reduction on proteobacteria till W12, while a major rebound of this text was observed in EEN at week 12. the reader can have a great idea about the differences between CDED and EEN.
Author's response: Thank you very much for the comment. More information about the differences between EEN and CDED has been added in the text (lines 259-268)
Reviewer 2 Report
This is a review on nutritional therapy in pediatric Crohn's disease, and more specifically on exclusion diet. The review is clear, and summarizes well previous reports.
I would be less enthusiastic about children tolerance. Children acceptance is not that good: patients get tired of the routine diet, they are not compliant after 1 or 2 weeks: eating every day the same finally disgusts them.
Moreover some products are from israel and not so easy to find elsewhere in winter.
Author Response
Comment 1: I would be less enthusiastic about children tolerance. Children acceptance is not that good: patients get tired of the routine diet, they are not compliant after 1 or 2 weeks: eating every day the same finally disgusts them.
Authors' answer: Thank you very much for this comment. Based on the results of a study by Professor Levine's team (publication in Gastroenterology, 2019), the tolerance of the CDED diet among children was high and amounted to as much as 97.5%. Of course, we agree that these results require confirmation in subsequent studies. Our own experience, based on a group of 70 patients, is also good. In the near future we are planning to summarize our results, also taking into account the diet's tolerance. Then it will be possible to relate to the oryginal results of the team from Israel.
Comment 2: Moreover some products are from israel and not so easy to find elsewhere in winter.
Authors' answer: Thank you very much for this comment. Of course, access to selected products based on where the pateint live can be a problem. Nevertheless, Prodesor Levine with team tried to choose the products so as to enable the use of the diet in various parts of the world. The basic products allowed in the first phase, i.e. chicken breast, eggs, bananas, potatoes, apples or rice, belong to the products available in many regions of the world. In addition, knowing the theoretical assumptions of the CDED diet, an experienced specialist can replace some products with others, while maintaining the basic principle of the diet, i.e. the exclusion of products with a potentially negative impact on the pathomechanics of Crohn's disease.
Round 2
Reviewer 1 Report
Nice improvement and good job. In line 304 there is a mistake, you should say taxa and not tax. please correct it , because it is very important terminology. except that, all fine
Author Response
Dear Reviewer,
Please find attached Word file with all answers.
Regards
Authors
